# Ethical Considerations for Responsible Data Curation

**Jerone T. A. Andrews**[*]
Sony AI, Tokyo

**Dora Zhao**[†]
Sony AI, New York

**William Thong**[†]
Sony AI, Zurich

**Apostolos Modas**[†]
Sony AI, Zurich

**Orestis Papakyriakopoulos**[†]
Sony AI, Zurich

**Alice Xiang**
Sony AI, Seattle

## Abstract

Human-centric computer vision (HCCV) data curation practices often neglect privacy and bias concerns, leading to dataset retractions and unfair models. HCCV datasets constructed through nonconsensual web scraping lack crucial metadata for comprehensive fairness and robustness evaluations. Current remedies are post hoc, lack persuasive justification for adoption, or fail to provide proper contextualization for appropriate application. Our research focuses on proactive, domain-specific recommendations, covering purpose, privacy and consent, and diversity, for curating HCCV evaluation datasets, addressing privacy and bias concerns. We adopt an ante hoc reflective perspective, drawing from current practices, guidelines, dataset withdrawals, and audits, to inform our considerations and recommendations.

## 1 Introduction

Contemporary human-centric computer vision (HCCV) data curation practices, which prioritize dataset size and utility, have pushed issues related to privacy and bias to the periphery, resulting in dataset retractions and modifications [78, 126, 175, 216, 244, 320], as well as models that are unfair or rely on spurious correlations [22, 26, 112, 139, 146, 215, 272, 281]. HCCV datasets primarily rely on nonconsensual web scraping [99, 122, 124, 228, 260, 266, 310]. These datasets not only regard image subjects as free raw material [32], but also lack the ground-truth metadata required for fairness and robustness evaluations [91, 171, 196, 216]. This makes it challenging to obtain a comprehensive understanding of model blindspots and cascading harms [30, 85] across dimensions, such as data subjects, instruments, and environments, which are known to influence performance [222]. While, for example, image subject attributes can be inferred [7, 43, 170, 188, 198, 241, 267, 290, 343, 375], this is controversial for social constructs, notably race and gender [28, 132, 179, 180]. Inference introduces further biases [19, 107, 147, 263, 291] and can induce psychological harm when incorrect [47, 275].

Recent efforts in machine learning (ML) to address these issues often rely on post hoc reflective processes. Dataset documentation focuses on interrogating and describing datasets after data collection [5, 27, 44, 94, 108, 149, 243, 258, 273, 307]. Similarly, initiatives by NeurIPS and ICML ask authors to consider the ethical and societal implications of their research after completion [257]. Further, dataset audits [247, 292] and bias detection tools [29, 340] expose dataset management issues and representational biases without offering guidance on responsible data collection. Although there are existing proposals for artificial intelligence (AI) and data design guidelines [35, 79, 116, 160, 202, 247], as well as calls to adopt methodologies from more established fields [154, 159, 166], general-purpose guidelines lack domain specificity and task-oriented guidance [307]. For example, remedies may prioritize privacy and governance [35] but overlook data composition and image content. Other recommended practices lack persuasive justification for adoption [116, 160] or fail to provide proper

---

[*]Correspondence to `jerone.andrews@sony.com`.
[†]Equal contribution; authors are listed in random order.

37th Conference on Neural Information Processing Systems (NeurIPS 2023) Track on Datasets and Benchmarks.

contextualization for appropriate application [252, 323, 367]. For instance, the *People + AI Guide-book* [116] suggests creating dataset specifications without explaining the rationale, and privacy methodologies are advocated without cognizant of privacy and data protection laws [252, 323, 367]. These efforts, which hold significance in promoting responsible practices, would benefit from being supplemented by proactive, domain-specific recommendations aimed at tackling privacy and bias concerns starting from the inception of a dataset.

Our research directly addresses these critical concerns by examining purpose (Section 3), consent and privacy (Section 4), and diversity (Section 5). Compared to recent scholarship, we adopt an ante hoc reflective perspective, offering considerations and recommendations for curating HCCV datasets for fairness and robustness evaluations. Our work, therefore, resonates with the call for domain-specific resources to operationalize fairness [68, 150, 305]. We draw insights from current practices [42, 170, 375], guidelines [31, 222, 231], dataset withdrawals [78, 126, 216], and audits [35, 36, 247], to motivate each recommendation, focusing on HCCV evaluation datasets that present unique challenges (e.g., visual leakage of personally identifiable information) and opportunities (e.g., leveraging image metadata for analysis). To guide curators towards more ethical yet resource-intensive curation, we provide a checklist in Appendix A.[3] This translates our considerations and recommendations into pre-curation questions, functioning as a catalyst for discussion and reflection.

While several of our recommendations can also be applied retroactively such measures cannot undo incurred harm, e.g., resulting from inappropriate uses, privacy violations, and unfair representation [129]. It is important to make clear that our proposals are not intended for the evaluation of HCCV systems that detect, predict, or label sensitive or objectionable attributes such as race, gender, sexual orientation, or disability.

## 2 Development Process

HCCV should adhere to the most stringent ethical standards to address privacy and bias concerns. As stated in the NeurIPS Code of Ethics [1], it is essential to abide by established institutional research protocols, ensuring the safeguarding of human subjects. These protocols, initially designed for biomedical research, have, however, been met with confusion, resulting in inconsistencies when applied in the context of data-centric research [218]. For example, HCCV research often amasses millions of "public" images without obtaining informed consent or participation, disregarding serious privacy and ethical concerns [3, 35, 133, 245, 301]. This exemption from research ethics regulation is grounded in the limited definition of human-subjects research, which categorizes extant, publicly available data as minimal risk [218, 256]. Thus, numerous ethically-dubious HCCV datasets would not fall under Institutional Review Board (IRB) oversight [247]. What's more, the NeurIPS Code of Ethics only mandates following existing protocols when research involves "direct" interaction between human participants and researchers or technical systems. Even when research is subjected to supervision, IRBs are restricted from considering broader societal consequences beyond the immediate study context [217]. Compounding matters, CV-centric conferences are still to adopt ethics review practices [306].

These limitations are concerning, especially considering the potential for predictive privacy harms when seemingly non-identifiable data is combined [35, 70, 218] or when data is used for harmful downstream applications such as predicting sexual orientation [192, 344], crime propensity [358, 365], or emotion [10, 224]. Acknowledging this, our research study employed the same principles underpinning established guidelines [24, 331] for protecting human subjects in research to identify ethical issues in HCCV dataset design, namely autonomy, justice, beneficence, and non-maleficence. *Autonomy* respects individuals' self-determination—e.g., through informed consent and assent for HCCV datasets. *Justice* promotes the fair distribution of risks, costs, and benefits, guiding decisions on compensation, data accessibility, and diversity. *Beneficence* entails the proactive promotion of positive outcomes and well-being, e.g., by soliciting individuals' to self-identify, while *non-maleficence* centers on minimizing harm and risks during dataset design, e.g., by redacting privacy-leaking image regions and metadata.

To ensure comprehensive consideration, we harnessed diverse expertise, following contemporary, interdisciplinary practices [261, 270, 307]. Our team comprises researchers, practitioners, and

---

[3]The checklist can also be found at: `https://github.com/SonyResearch/responsible_data_curation`.

lawyers with backgrounds in ML, CV, algorithmic fairness, philosophy, and social science. With a range of ethnic, cultural, and gender backgrounds, we bring extensive experience in designing CV datasets, training models, and developing ethical guidelines. To align our expertise with the principles, we collectively discussed them, considering each author's background. After identifying key ethical issues in HCCV data curation practices, we iteratively refined them into an initial draft of ethical considerations. We extensively collected, analyzed, and discussed papers spanning a range of themes such as HCAI, HCCV datasets, data and model documentation, bias detection and mitigation, AI and data design, fairness, and critical AI. Our comprehensive literature review incorporated pertinent studies and datasets, resulting in refined considerations with detailed explanations and recommendations for responsible data curation. Additional details are provided in Appendix B.

## 3 Purpose

In ML, significant emphasis has been placed on the acquisition and utilization of "general-purpose" datasets [259]. Nevertheless, without a clearly defined task pre-data collection, it becomes challenging to effectively handle issues related to data composition, labeling, data collection methodologies, informed consent, and assessments related to data protection. This section addresses conflicting dataset motivations and provides recommendations.

### 3.1 Ethical Considerations

**Fairness-unaware datasets are inadequate for measuring fairness.** Datasets lacking explicit fairness considerations are inadequate for mitigating or studying bias, as they often lack the necessary labels for assessing fairness. For instance, the COCO dataset [196], focused on scene understanding, lacks subject information, making fairness assessments challenging. Researchers, consequently, resort to human annotators to infer, e.g., subject characteristics, limiting bias measurement to visually "inferable" attributes. This introduces annotation bias [56] and the potential for harmful inferences [47, 275].

**Fairness-aware datasets are incompatible with common HCCV tasks.** Industry practitioners stress the importance of carefully designed and collected "fairness-aware" datasets to detect bias issues [150]. Fabris et al. [93] found that out of 28 CV datasets used in fairness research between 2014 and 2021, only eight were specifically created with fairness in mind. Among these, seven were HCCV datasets (scraped from the web) [43, 170, 216, 308, 319, 342, 343], including five focused on facial analysis. Due to the limited availability and delimited task focus of fairness-aware datasets, researchers repurpose "fairness-unaware" datasets [120, 139, 196, 198, 208, 346, 373]. Fairness-aware datasets fall short in addressing the original tasks associated with well-known HCCV datasets, which encompass a range of tasks, such as segmentation [64, 209], pose estimation [13, 196], localization and detection [73, 91, 110], identity verification [153], action recognition [173], as well as reconstruction, synthesis and manipulation [114, 171]. The absence of fairness-aware datasets with task-specific labels hampers the practical evaluation of HCCV systems, despite their importance in domains such as healthcare [155, 220], autonomous vehicles [163], and sports [317]. Additionally, fairness-aware datasets lack self-identified annotations from image subjects, relying on inferred attributes, e.g., from online resources [43, 308, 319].

### 3.2 Practical Recommendations

**Refrain from repurposing datasets.** Existing datasets, repurposable but optimized for specific functions, can inadvertently perpetuate biases and undermine fairness [183]. Repurposing fairness-unaware data for fairness evaluations can result in *dirty data*, characterized by missing or incorrect information and distorted by individual and societal biases [181, 265]. Dirty data, including inferred data, can have significant downstream consequences, compromising the validity of research, policy, and decision-making [14, 63, 265, 341]. ML practitioners widely agree that a proactive approach to fairness is preferable, involving the direct collection of demographic information from the outset [150]. To mitigate epistemic risk, curated datasets should capture key dimensions influencing fairness and robustness evaluation of HCCV models, i.e., data subjects, instruments, and environments. Model Cards explicitly highlight the significance of these dimensions in fairness and robustness assessments [222].

**Create purpose statements.** Pre-data collection, dataset creators should establish *purpose statements*, focusing on motivation rather than cause [129]. Purpose statements address, e.g., data collection motivation, desired composition, permissible uses, and intended consumers. While dataset documentation [108, 258] covers similar questions, it is a *reflective* process and can be manipulated to fit the narrative of the collected data, as opposed to directing the narrative of the data to be collected. Purpose statements can play a crucial role in preventing both *hindsight bias* [51, 97, 176] and *purpose creep*, ensuring alignment with stakeholders' consent and intentions [186]. To enhance transparency and accountability, as recommended by Peng et al. [247], purpose statements can undergo peer review, similar to *registered reports* [238]. Registered reports, recognized by the UK 2021 Research Excellence Framework, incentivize rigorous research practices and can lead to increased institutional funding [51].

## 4    Consent and Privacy

Informed consent is crucial in research ethics involving humans [230, 235], ensuring participant safety, protection, and research integrity [59, 253]. Shaping data collection practices in various fields [35, 235], informed consent consists of three elements: *information* (i.e., the participant should have sufficient knowledge about the study to make their decision), *comprehension* (i.e., the information about the study should be conveyed in an understandable manner), and *voluntariness* (i.e., consent must be given free of coercion or undue influence). While consent is not the only legal basis for data processing, it is globally preferred for its legitimacy and ability to foster trust [82, 253]. We address concerns related to consent and privacy, and provide recommendations.

### 4.1    Ethical Considerations

**Human-subjects research.**  As aforementioned in Section 2, HCCV datasets are frequently collected without informed consent or participation, primarily due to the classification of publicly available data as "minimal risk" within human-subjects research. However, beyond possible predictive privacy harms and unethical downstream uses, collecting data without informed consent hinders researchers and practitioners from fully understanding and addressing potential harms to data subjects [218, 333]. Some argue that consent is pivotal as it provides individuals with a last line of defense against the misuse of their personal information, particularly when it contradicts their interests or well-being [77, 223, 245, 253].

**Creative Commons loophole.**  Some datasets have been created based on the misconception that the "unlocking [of] restrictive copyright" [35] through Creative Commons licenses implies data subject consent. However, the Illinois Biometric Information Privacy Act (BIPA) [161] mandates data subject consent, even for publicly available images [370]. In the UK and EU General Data Protection Regulation (GDPR) [88] Article 4(11), images containing faces are considered biometric data, requiring "freely given, specific, informed, and unambiguous" consent from data subjects for data processing. Similarly, in China, the Personal Information Protection Law (PIPL) [233] Article 29 mandates obtaining individual consent for processing sensitive personal information, including biometric data (Article 28). While a Creative Commons license may release copyright restrictions on specific artistic expressions within images [370], it does not apply to image regions containing biometric data such as faces, which are protected by privacy and data protection laws [300].

**Vulnerable persons.**  Nonconsensual data collection methods can result in the inclusion of vulnerable individuals unable to consent or oppose data processing due to power imbalances, limited capacity, or increased risks of harm [89, 207]. While scraping vulnerable individuals' biometric data may be incidental, some researchers actively target them, jeopardizing their sensitive information without guardian consent [128, 260].

Paradoxically, attempts to address racial bias in data have involved soliciting homeless persons of color, further compromising their vulnerability [103]. When participation is due to economic or situational vulnerability, as opposed to one's best interests, monetary offerings may be perceived as inducement [117]. Further ethical concerns manifest when it is unclear whether participants were adequately *informed* about a research study. For instance, in ethnicity recognition research [72], despite obtaining informed consent, criticism arose due to training a model that discriminates between Chinese Uyghur, Korean, and Tibetan faces. Although the study's focus is on the technology

itself [315], its potential use in enhancing surveillance on Chinese Uyghurs raises ethical questions due to the human rights violations against them [333].

**Consent revocation.** Dataset creators sometimes view autonomy as a challenge to collecting biometric data for HCCV, especially when data subjects prioritize privacy [214, 287, 297]. Nonetheless, informed consent emphasizes *voluntariness*, encompassing both the ability to give consent and the right to withdraw it at any time [74]. GDPR grants explicit revocation rights (Article 7) and the right to request erasure of personal data (Article 17) [350]. However, image subjects whose data is collected without consent are denied these rights. The nonconsensual FFHQ face dataset [171] offers an opt-out mechanism, but since inclusion was *involuntary*, subjects may be unaware of their inclusion, rendering the revocation option hollow. Moreover, this burdens data subjects with tracking the usage of their data in datasets, primarily accessible by approved researchers [81].

**Image- and metadata-level privacy attributes.** Researchers have focused on obfuscation techniques, e.g., blurring, inpainting, and overlaying, to reduce private information leakage of nonconsensual individuals [46, 101, 194, 195, 213, 252, 311, 323, 362, 367]. Nonetheless, face detection algorithms used in obfuscation may raise legal concerns, particularly if they involve predicting facial landmarks, potentially violating BIPA [61, 370]. BIPA focuses on collecting and using face geometry scans regardless of identification capability, while GDPR protects any identifiable person, requiring data holders to safeguard the privacy of nonconsenting individuals. Moreover, reliance on automated face detection methods raises ethical concerns, as demonstrated by the higher precision of pedestrian detection models on lighter skin types compared to darker skin types [352]. This predictive inequity leads to allocative harm, denying certain groups opportunities and resources, including the rights to safety [322] and privacy [80].

It is important to note that face obfuscation may not guarantee privacy [145, 367]. The Visual Redactions dataset [242] includes 68 image-level privacy attributes, covering biometrics, sensitive attributes, tattoos, national identifiers, signatures, and contact information. Training faceless person recognition systems using full-body cues reveals higher than chance re-identification rates for face blurring and overlaying [239], indicating that solely obfuscating face regions might be insufficient under GDPR. Furthermore, image metadata can also disclose sensitive details, e.g., date, time, and location, as well as copyright information that may include names [11, 239]. This is worrisome for users of commonly targeted platforms like Flickr, which retain metadata by default.

## 4.2 Practical Recommendations

**Obtain voluntary informed consent.** Similar to recent consent-driven HCCV datasets [136, 254, 268], explicit informed consent should be obtained from each person depicted in, or otherwise identifiable, in a dataset, allowing the sharing of their facial, body, and biometric information for evaluating the fairness and robustness of HCCV technologies. Datasets collected with consent *reduce* the risk of being fractured, however, data subjects may later revoke their consent over, e.g., privacy concerns they may not have been aware of at the time of providing consent or language nuances [65, 379]. Following GDPR (Article 7), plain language consent and notice forms are recommended to address the lack of public understanding of AI technologies [199].

When collecting images of individuals under the age of majority or those whose ability to protect themselves is significantly impaired on account of disability, illness, or otherwise, guardian consent is necessary [182]. However, relying solely on guardian consent overlooks the views and dignity of the vulnerable person [141]. To address this, in addition to guardian consent, voluntary informed *assent* can be sought from a vulnerable person, in accordance with UNICEF's principlism-guided data collection procedures [31, 327]. When employing appropriate language and tools, assent establishes the vulnerable person understands the use of their data and willingly participates [31]. If a vulnerable person expresses dissent or unwillingness to participate, their data should not be collected, regardless of guardian wishes.

Informed by the U.S. National Bioethics Advisory Commission's *contextual vulnerability framework* [60], dataset creators should assess vulnerability on a continuous scale. That is, the circumstances of participation should be considered, which may require, e.g., a participatory design approach, assurances over compensation, supplementary educational materials, and insulation from hierarchical or authoritative systems [117].

**Adopt techniques for consent revocation.** To permit consent revocation, dataset creators should implement an appropriate mechanism. One option is *dynamic consent*, where personalized communication interfaces enable participants to engage more actively in research activities [174, 348]. This approach has been implemented successfully through online platforms, offering options for blanket consent, case-by-case selection, or opt-in depending on the data's use [174, 211, 314]. Alternatively, another recommended approach is to establish a steering board or charitable trust composed of representative dataset participants to make decisions regarding data use [255]. The feasibility of these proposals may vary based on a dataset's scale. Nonetheless, at a minimum, data subjects should be provided a simple and easily accessible method to revoke consent [136, 254, 268]. This aligns with guidance provided by the UK Information Commission's Office (ICO), emphasizing the need to provide alternatives to online-based revocation processes to accommodate varying levels of technology competency and internet access among data subjects [325].

**Collect country of residence information.** Anonymizing nonconsensual human subjects through face obfuscation, as done in datasets such as ImageNet [367], may not respect the privacy laws specific to the subjects' country of residence. To comply with relevant data protection laws, dataset curators should collect the country of residence from each data subject to determine their legal obligations, helping to ensure that data subjects' rights are protected and future legislative changes are addressed [249, 268]. For instance, GDPR Article 7(3) grants data subjects the right to withdraw consent at any time, which was not explicitly addressed in its predecessor [253].

**Redact privacy leaking image regions and metadata.** The European Data Protection Board emphasizes that anonymization of personal data must guard against re-identification risks such as singling out, linkability, and inference [76]. Re-identification remains possible even when nonconsensual subjects' faces are obfuscated, through other body parts or contextual information [242]. One solution is to redact all privacy-leaking regions related to nonconsensual subjects (including their entire bodies, clothing, and accessories) and text (excluding copyright owner information). However, anonymization approaches should be validated empirically, especially when using methods without formal privacy guarantees. Moreover, to mitigate algorithmic failures or biases, human annotators should be involved in creating region proposals, as well as verifying automatically generated proposals, for image regions with identifying or private information [367]. For nonconsensual individuals residing in certain jurisdictions (e.g., Illinois, California, Washington, Texas), automated region proposals requiring biometric identifiers should be avoided. Instead, human annotators should take the responsibility of generating these proposals.

Notwithstanding, to further protect privacy, dataset creators should take steps to ensure that image metadata does not reveal identifying information that data subjects did not consent to sharing. This may involve replacing exact geolocation data with a more general representation, such as city and country, and excluding user-contributed details from metatags containing personally identifiable information, except when this action would violate copyright. However, we do not advise blanket redaction of all metadata, as it contains valuable image capture information that can be useful for assessing model bias and robustness related to instrument factors.

## 5 Diversity

HCCV dataset creators widely acknowledge the significance of dataset diversity [13, 64, 78, 170, 171, 173, 196, 283, 361, 368], realism [110, 153, 164, 173, 196, 368], and difficulty [13, 16, 64, 73, 78, 91, 110, 173, 196, 198, 361, 368] to enhance fairness and robustness in real-world applications. Previous research has emphasized diversity across image subjects, environments, and instruments [43, 139, 222, 287], but there are many ethical complexities involved in specifying diversity criteria [14, 15]. This section examines taxonomy challenges and offers recommendations.

### 5.1 Ethical Considerations

**Representational and historical biases.** The Council of Europe have expressed concerns about the threat posed by AI systems to equality and non-discrimination principles [67]. Many dataset creators often prioritize protected attributes, i.e., gender, race, and age, as key factors of dataset diversity [287]. Nevertheless, most HCCV datasets exhibit historical and representational biases [35, 166, 172, 312, 366]. These biases can be pernicious, particularly when models learn and *amplify* them. For instance, image captioning models may rely on contextual cues related to activities like shopping [377]

and laundry [376] to generate gendered captions. Spurious correlations are detrimental, as they are not causally related and perpetuate harmful associations [112, 281]. In addition, prominent examples in HCCV research demonstrate disparate algorithmic performance based on race and skin color [42, 43, 54, 123, 142–144, 148, 250, 271, 299, 318, 334, 375]. Most recently, autonomous robots have displayed racist, sexist, and physiognomic stereotypes [158]. Furthermore, face detection models have shown lower accuracy when processing images of older individuals compared to younger individuals [369]. While not endorsing these applications, discrepancies have also been observed in facial emotion recognition services for children in both commercial and research systems [152, 363], as well as age estimation [58, 115, 200].

Despite concerns regarding privacy, liability, and public relations, the collection of special and sensitive category data is crucial for bias assessments [15]. GDPR guidance from the UK ICO confirms that sensitive attributes can be collected for fairness purposes [324]. However, obtaining this information presents challenges, such as historical mistrust in clinical research among African-Americans [92, 191] or the social stigma of being photographed that some women face [166]. Nonetheless, marginalized communities may require explicit explanations and assurances about data usage to address concerns related to service provision, security, allocation, and representation [359]. This is particularly important as remaining *unseen* does not protect against being *mis-seen* [359].

**The digital divide and accessibility.** Healthcare datasets often lack representation of minority populations, compromising the reliability of automated decisions [356]. The World Health Organization (WHO) emphasizes the need for data accuracy, completeness, and diversity, particularly regarding age, in order to address ageism in AI [355]. ML systems may prioritize younger populations for resource allocation, assuming they would benefit the most in terms of life expectancy [355]. The digital divide further exacerbates the underrepresentation of vulnerable groups, including older generations, low-income school-aged children, and children in East Asia and the South Pacific who lack access to digital technology [326, 354]. Insufficient access to digital technology hampers the representation of vulnerable persons in datasets [290], leading to *outcome homogenization*—i.e., the systematic failing of the same individuals or groups [39].

**Confused taxonomies.** Sex and gender are often used interchangeably, treating gender as a consequence of one's assigned sex at birth [95]. However, this approach erases intersex individuals who possess non-binary physiological sex characteristics [95]. Treating sex and gender as interchangeable perpetuates normative views by casting gender as binary, immutable, and solely based on biological sex [179]. This perspective disregards transgender and gender nonconforming individuals. Moreover, sex, like gender, is a social construct, as sexed bodies do not exist outside of their *social context* [45].

Similar to sex and gender, race and ethnicity are often used synonymously [332]. Nations employ diverse census questions to ascertain ethnic group composition, encompassing factors such as nationality, race, color, language, religion, customs, and tribe [328]. However, these categories and their definitions lack consistency over time and geography, often influenced by political agendas and socio-cultural shifts [286]. This variability makes it challenging to collect globally representative and meaningful data on ethnic groups. Consequently, several HCCV datasets have incorporated inconsistent and arbitrary racial categorization systems [7, 267, 343, 374]. For instance, the FairFace dataset [170] creators reference the US Census Bureau's racial categories without considering the social definition of race they represent [240]. The US Census Bureau explicitly states that their categories reflect a social definition rather than a biological, anthropological, or genetic one. Consequently, labeling the "physical race" of image subjects based on nonphysiological categories is contradictory. Furthermore, the FairFace creators do not disclose the demographics or cultural compatibility of their annotators.

**Own-anchor bias.**     HCCV approaches for encoding age in datasets vary, using either integer labels [53, 102, 125, 226, 236, 264, 277, 278, 374] or group labels [84, 105, 193, 302]. Age groupings are often preferred when collecting unconstrained images from the web, as human annotators must infer subjects' ages, which is challenging [48]. This is evident in crowdsourced annotations, where 40.2% of individuals in the OpenImages MIAP dataset [290] could not be categorized into an age group. Factors unrelated to age, such as facial expression [106, 237, 345] and makeup [83, 237, 313], influence age perception. Furthermore, annotators have exhibited lower accuracy when labeling people outside of their own demographic group [8, 9, 113, 279, 303, 335, 339].

**Post hoc rationalization of the use of physiological markers.**     Gender information about data subjects is obtained through inference [53, 170, 187, 198, 236, 264, 277, 278, 290, 343, 374, 374]

or self-identification [136, 190, 203, 204, 375]. Inference raises concerns as it assumes that gender can be determined solely from imagery without consent or consultation with the subject, which is noninclusive and harmful [87, 127, 179]. Even when combined with non-image-based information, inferred gender fails to account for the fluidity of identity, potentially mislabeling subjects at the time of image capture [277, 278]. Moreover, physical traits are just one of many dimensions, including posture, clothing, and vocal cues, used to infer not only gender but also race [100, 177, 179].

**Erasure of nonstereotypical individuals.** HCCV datasets frequently adopt a US-based racial schema [170, 190, 203, 204, 264, 343], which may oversimplify and essentialize groups [316]. This approach may not align with other more nuanced models, e.g., the continuum-based color system used in Brazil, which considers a wide range of physical characteristics. Nonconsensual image datasets rely on annotators to assign semantic categories, perpetuating stereotypes and disseminating them beyond their cultural context [180]. Notably, images without label consensus are often discarded [170, 267, 343], potentially excluding individuals who defy stereotypes, such as multi-ethnic individuals [276].

**Phenotypic attributes.** Protected attributes may not be the most appropriate criteria for evaluating HCCV models [43]. Social constructs like race and gender lack clear delineations for subgroup membership based on visible or invisible characteristics. These labels capture invisible aspects of identity that are not solely determined by visible appearance. Moreover, the phenotypic characteristics within and across subgroups exhibit significant variability [25, 48, 96, 138, 180, 347].

**Environment and instrument.** The image capture device and environmental conditions significantly influence model performance, and their impact should be considered [222]. Factors such as camera software, hardware, and environmental conditions affect HCCV model robustness in various settings [4, 140, 197, 221, 229, 353, 360, 364, 371]. Understanding performance differences is crucial from ethical and scientific perspectives. For example, sensitivity to illumination or white balance may be linked to sensitive attributes, e.g., skin tone [62, 184, 185, 378], while available instruments or environmental co-occurrences may correlate with demographic attributes [139, 295, 376].

**Annotator positionality.** Psychological research highlights the influence of annotators' sociocultural background on their visual perception [19, 107, 147, 263, 275, 291]. However, recent empirical studies have evidenced a lack of regard for the impact an annotator's social identity has on data [79, 111]. Only a handful of HCCV datasets provide annotator demographic details [12, 56, 287, 375].

**Recruitment and compensation.** Data collected without consent patently lacks compensation. Balancing between excessive and deficient payment is crucial to avoid coercion and exploitation [231, 268]. An additional concern is the employment of remote workers from disadvantaged regions [248], often with low wages and fast-paced work conditions [71, 135, 162, 206]. This can lead to arbitrary denial of payment based on opaque quality criteria [98] and prevents union formation [206], creating a sense of invisibility and uncertainty for workers [321].

## 5.2 Practical Recommendations

**Obtain self-reported annotations.** Practitioners are cautious about inferring labels about people to avoid biases [15]. Moreover, data access request rights, e.g., as offered by GDPR, California Consumer Privacy Act, and PIPL, may require data holders to disclose inferred information. To avoid stereotypical annotations and minimize harm from misclassification [275], labels should be collected directly from image subjects, who inherently possess contextual knowledge of their environment and awareness of their own attributes.

**Provide open-ended response options.** Closed-ended questions, such as those on census forms, may lead to incongruous responses and inadequate options for self-identification [156, 179, 274]. Open-ended questions provide more accurate answers but can be taxing, require extensive coding, and are harder to analyze [40, 109, 178, 298]. To balance this, closed-ended questions should be augmented with an open-ended response option, avoiding the term "other", which implies *othering* norms [285]. This gives subjects a *voice* [234, 296] and allows for future question design improvement.

**Acknowledge the mutability and multiplicity of identity.** *Identity shift*—the intentional self-transformation in mediated contexts [49]—is often overlooked. To address this, we propose collecting self-identified information on a per-image basis, acknowledging that identity is temporal and nonstatic. Specifically, for sensitive attributes, allowing the selection of multiple identity categories without limitations is preferable [304, 309]. This prevents oversimplification and marginalization. While we

acknowledge the potential burden of self-identification on fluid and dynamic identities, an image captures a single moment. Thus, evolving identity may not require metadata updates; however, we recommend providing subjects with mechanisms for updates when needed.

**Collect age, pronouns, and ancestry.** First, to capture accurate age information, dataset curators should collect the exact biological age in years from image subjects, corresponding to their age at the time of image capture. This approach offers flexibility, insofar as permitting the appropriate aggregation of the collected data. This is particularly important given the lack of consistent age groupings in the literature.

Second, dataset curators should consider opting to collect self-identified pronouns. This promotes mutual respect and common courtesy, reducing the likelihood of causing harm through misgendering [157]. Self-identified pronouns are particularly important for sexual and gender minority communities as they "convey and affirm gender identity" [232]. Significantly, pronoun use is increasingly prevalent in social media platforms [86, 165, 167], workplaces [55], and education settings [20, 212], fostering gender inclusivity [21]. However, subjects should always have the option of not disclosing this information.

Finally, to address issues with ethnic and racial classification systems [180, 286], dataset creators should consider collecting ancestry information instead. Ancestry is defined by historically shaped borders and has been shown to offer a more stable and less confusing concept [17]. The United Nations' M49 geoscheme can be used to operationalize ancestry [329], where subjects select regions that best describe their ancestry. To situate responses, subjects could be asked, e.g., "Where do your ancestors (e.g., great-grandparents) come from?". This avoids reliance on proxies, e.g., skin tone, that risk normalizing their inadequacies without reflecting their limitations [15].

**Collect aggregate data for commonly ignored groups.** Additional sensitive attributes should also be collected, such as disability and pregnancy status, when voluntarily disclosed by subjects. These attributes should be reported in aggregate data to reduce the safety concerns of subjects [309, 351]. Given that definitions of these attributes may be inconsistent and tied to culture, identity, and histories of oppression [37, 41], navigating tensions between benefits and risks is necessary. Despite potential reluctance, sourcing data from underrepresented communities contributes to dataset inclusivity [37, 168]. Regarding disability, the American Community Survey [330] covers categories related to hearing, vision, cognitive, ambulatory, self-care, and independent living difficulties.

**Collect phenotypic and neutral performative features.** Collecting phenotypic characteristics can serve as *objective* measures of diversity, i.e., attributes which, in evolutionary terms, contribute to individual-level recognition [57], e.g., skin color, eye color, hair type, hair color, height, and weight [19]. These attributes have enabled finer-grained analysis of model performance and biases [43, 75, 294, 318, 349, 372]. Additionally, considering a multiplicity of *neutral performative features*, e.g., facial hair, hairstyle, cosmetics, clothing, and accessories, is important to surface the perpetuation of social stereotypes and spurious relationships in trained models [6, 18, 166, 284, 340].

**Record environment and instrument information.** Data should capture variations in environmental conditions and imaging devices, including factors such as image capture time, season, weather, ambient lighting, scene, geography, camera position, distance, lens, sensor, stabilization, use of flash, and post-processing software. Instrument-related factors may be easily captured, by restricting data collection to images with exchangeable image file format (Exif) metadata. The remaining factors, e.g., season and weather can be self-reported or coarsely estimated utilizing information such as image capture time and location.

**Recontextualize annotators as contributors.** Dataset creators should document the identities of annotators and their contributions to the dataset [12, 79], rather than treating them as anonymous entities responsible for data labeling alone [52, 206]. While many datasets [78, 137, 196] neglect to report annotator demographics, assuming objectivity in annotation for visual categories is flawed [23, 169, 219]. Furthermore, using majority voting to reach the assumed ground truth, disregards minority opinions, treating them as noise [169]. Annotator characteristics, including pronouns, age, and ancestry, should be recorded and reported to quantify and address annotator perspectives and bias in datasets [12, 118]. Additionally, allowing annotators freedom in labeling helps to avoid replicating socially dominant viewpoints [219].

**Fair treatment and compensation for contributors.** In accordance with Australia's National Health and Medical Research Council [231] and the WHO [66], dataset contributors should not only be

guaranteed compensation above the minimum hourly wage of their country of residence [280], but also according to the complexity of tasks to be performed. However, alternative payment models, for example, based on the average hourly wage, may offer benefits in terms of promoting diversity by increasing the likelihood of higher socio-economic status contributors [251].

Besides payment, the implementation of direct communication channels and feedback mechanisms, such as anonymized feedback forms [246], can help to address issues faced by annotators while providing a level of protection from retribution. Furthermore, the creation of plain language guides can ease task completion and reduce quality control overheads. Ideally, recruitment and compensation processes should be well-documented and undergo ethics review, which can help to further reduce the number of "glaring ethical lapses" [293].

# 6   Discussion and Conclusion

Supplementary to established ethical review protocols, we have provided proactive, domain-specific recommendations for curating HCCV evaluation datasets for fairness and robustness evaluations. However, encouraging change in ethical practice could encounter resistance or slow adoption due to established norms [218], inertia [33], diffusion of responsibility [151], and liability concerns [15]. To garner greater acceptance, platforms such as NeurIPS could adopt a model similar to the registered reports format, embraced by over 300 journals [51, 247]. This entails pre-acceptance of dataset proposals before curation, alleviating financial uncertainties associated with more ethical practices.

Nevertheless, seeking consent from all depicted individuals might give rise to logistical challenges. Resource requirements tied to the implementation and maintenance of consent management systems could emerge, potentially necessitating significant investment in technical infrastructure and dedicated personnel. Particularly for smaller organizations and academic research groups, these limitations could present considerable hurdles. A potential solution is forming *data consortia* [121, 166], which helps address operational challenges by pooling resources and knowledge.

Extending our recommendations to the curation of "democratizing" foundation model-sized training datasets [104, 119, 288, 289] poses an economic challenge. To put this into perspective, the GeoDE dataset of 62K crowdsourced object-centric images [262], without personally identifiable information, incurred a cost of $1.08 per image. While our recommendations may not seamlessly *scale* to the curation of fairness-aware, billion-sized image datasets, it is worth considering that "solutions which resist scale thinking are necessary to undo the social structures which lie at the heart of social inequality" [130]. Large-scale, nonconsensual datasets driven by scale thinking have included harmful and distressing content, including rape [35, 36], racist stereotypes [131], vulnerable persons [128], and derogatory taxonomies [34, 69, 129, 183]. Such content may further generate legal concerns [2]. We contend that these issues can be mitigated through the implementation of our recommendations.

Nonetheless, balancing resources between model development and data curation is value-laden, shaped by "social, political, and ethical values" [38]. While organizations readily invest significantly in model training [227, 336], compensation for data contributors often appears neglected [337, 338], disregarding that "most data represent or impact people" [380]. Remedial actions could be envisioned to bridge the gap between models developed with ethically curated data and those benefiting from expansive, nonconsensually crawled data. Reallocating research funds away from dominant data-hungry methods [38] would help to strike a balance between technological advancement and ethical imperatives.

However, the granularity and comprehensiveness of our diversity recommendations could be adapted beyond evaluation contexts, particularly when employing "fairness without demographics" [50, 134, 189, 210] training approaches, reducing financial costs. Nevertheless, the applicability of any proposed recommendation is intrinsically linked to the specific context [243]. Decisions should be guided by the social framework of a given application to ensure ethical and equitable data curation.

Just as the concepts of identity evolve, our recommendations must also evolve to ensure their ongoing relevance and sensitivity. Thus, we encourage dataset creators to tailor our recommendations to their *context*, fostering further discussions on responsible data curation.

## Acknowledgments and Disclosure of Funding

This work was funded by Sony Research.

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

# A  Responsible Data Curation Checklist for Fairness and Robustness Evaluations

This checklist translates our HCCV data curation considerations and recommendations into action items for researchers and practitioners. Presented as a series of questions, these items are designed to stimulate discussions among data collection teams. The questions are purposefully worded to avoid binary responses, encouraging open-ended dialogues. The primary focus of the checklist is to underscore the ethical dimensions and ensure that teams address concerns encompassing purpose, consent and privacy, as well as diversity.

It is important to engage with the checklist as a preliminary exercise before beginning data collection. This approach promotes informed decision-making and minimizes risks, leading to more responsible and reliable outcomes.

Contextual diversity is acknowledged to avoid a one-size-fits-all approach. Moreover, customization is encouraged, as not all items apply universally; teams should modify or expand the checklist to align with their context and use case. As with existing AI ethics checklists [90, 201, 205, 225, 269, 357], it is important to recognize that the checklist is not a guarantee for ethical compliance; rather, it functions as a catalyst for discussion and reflection.

We understand that answering these questions is time-consuming, increasing the burden on data collection teams whose work is already undervalued [247, 282]. Therefore, when navigating through these lists, priority should be put on items related to the specific domain and task of interest. The level of engagement needed for each question will invariably differ. Keep in mind that the questions aim to spur ethical thinking during dataset development: "Ethics is often about finding a good or better, but not perfect, answer" [380].

## A.1  Purpose

The questions in this section focus on eliciting strategies for curating HCCV evaluation datasets specifically for fairness and robustness assessments. They seek alignment with objectives and inquire about factors known to influence these assessments to ensure comprehensive evaluations. Moreover, the questions aim to assist in formulating clear dataset purpose statements, preventing ambiguity and misuse of data, as well as exploring external validation to enhance transparency and accountability.

**Dataset Development Strategy**

- Can you provide details about your strategy for developing a new dataset tailored specifically for conducting fairness and robustness assessments in the context of HCCV? How do you plan to ensure that this dataset is aligned with the objectives of evaluating fairness and robustness?

- Can you elaborate on the factors your dataset will encompass to comprehensively enable fairness and robustness evaluations for HCCV models? How do you intend to capture the primary factors, including data subjects, instruments, and environments, that influence these evaluations?

**Dataset Purpose Statement**

- Can you provide details about your plan to formulate a comprehensive dataset purpose statement? How will this statement effectively communicate the core motivations driving, e.g., data collection, outline the intended dataset composition, specify permissible uses of the data, and identify the specific audience you aim to serve with the dataset?

- Can you elaborate on your strategy for ensuring the accuracy and ethical alignment of your dataset's purpose statement? How do you plan to externally validate the content and ethical considerations of the statement?

- Can you provide insights into the benefits and implications of submitting your dataset's purpose statement as part of a research study proposal in the format of a registered report for your project?

## A.2 Consent and Privacy

The questions in this section explore informed consent, legal compliance, and privacy protection measures within anonymization strategies. The questions emphasize clarity and voluntariness in consent processes to prevent coercion or misuse of data. Moreover, they attempt to elicit strategies for explaining data collection purposes, consent revocation, and accommodating diverse participation circumstances. Furthermore, the questions seek insights into addressing anonymization challenges, aiming to prevent re-identification risks, unauthorized exposure, and legal noncompliance, while preserving data utility and protecting data subjects' rights.

### Informed and Voluntary Consent

- Can you elaborate on your approach to ensuring that you secure explicit, voluntary, and informed consent from all individuals who either appear in the dataset or can be discerned from it? How do you plan to handle consent for data annotators who may have disclosed personal information for the purposes of quantifying and addressing annotator perspectives and bias?

- Can you provide a comprehensive explanation of your strategy for conveying the purpose of data collection to the subjects? How do you intend to emphasize the utilization of their data, which includes various types of information such as facial, body, biometric images, as well as information about themselves and their environment, all in the context of assessing the fairness and robustness of HCCV systems?

- In what ways will you incorporate consent forms that are composed in plain language to enhance the understanding of AI technologies? How do you plan to make sure these forms effectively convey the intricacies of data usage?

- How do you plan to inform data subjects about their ability to withdraw consent at any given point during, or after, the data collection process? Can you provide details about the mechanisms you will have in place for facilitating this?

- Please provide insight into your strategy for collecting data from individuals below the age of majority or vulnerable individuals. How will you seek both guardian consent and voluntary informed assent in such cases?

- How do you plan to evaluate vulnerability along a continuous spectrum, taking into account contextual factors and recognizing that vulnerability is not solely binary or based solely on group affiliations, but can also be influenced by specific situations or circumstances?

- Can you also provide details about how you will consider the circumstances of participation, which might include the potential need for participatory design, assurances of compensation, provision of educational materials, and safeguards against authoritative structures? How will you address these various aspects in your approach?

- How do you intend to ensure that vulnerable individuals have a comprehensive understanding of the data usage and willingly provide informed assent? Can you outline the specific measures you intend to implement for this purpose?

- Can you elaborate on how you will respect the decision of a vulnerable individual who expresses dissent, regardless of the preferences of their guardian?

### Consent Revocation Mechanisms

- How do you plan to integrate mechanisms that allow data subjects to easily withdraw their consent? Can you provide specifics on how this process will be designed and executed?

- Can you provide insights into the benefits and implications of implementing dynamic consent mechanisms that utilize personalized communication interfaces? How do you intend to ensure that these mechanisms adapt to the preferences and needs of individual data subjects?

- How do you intend to enable data subjects to actively participate in research activities and manage their consent preferences? Can you provide more details about the tools or processes you plan to put in place to achieve this?

- In what ways will you explore the feasibility of online platforms for consent management that are user-friendly and minimize complexity for data subjects? What steps will you take to ensure easy accessibility?

- Can you provide insights into the options you will provide to data subjects for granting consent? How will you offer choices between blanket consent, case-by-case selection, or opt-in based on specific data usage?

- Can you elaborate on your considerations regarding the formation of a steering board or charitable trust composed of representative subjects from the dataset? How do you envision this entity contributing to decision-making processes?

- How do you plan to empower data subjects to actively participate in decisions concerning the usage of their data? What mechanisms or channels will you establish to facilitate this involvement?

- Can you provide information about the method you will offer data subjects to easily and promptly revoke their consent? How will you ensure that this process is straightforward and accessible?

- How do you intend to address varying levels of technological know-how and internet access among data subjects? Can you detail the measures you will take to accommodate these variations?

- What alternatives do you plan to offer for revoking consent that do not rely solely on online-based processes? How will you ensure that individuals with different needs and preferences can effectively revoke their consent?

- How do you plan to assess the practicality and suitability of the chosen mechanisms for consent revocation, taking into account the expected dataset size and the resources available to you? What criteria will you use to evaluate their effectiveness?

**Country of Residence Information**

- How do you plan to address the fact that anonymization measures might not universally meet legal requirements in specific regions, necessitating additional considerations? Can you provide insights into your strategy for ensuring legal compliance while implementing anonymization?

- Can you elaborate on your approach to collecting information about the country of residence for each individual in your dataset? How do you intend to use this information to ensure legal compliance and address potential privacy concerns?

- How do you plan to familiarize yourself with the data protection laws that are applicable in the countries of residence of your data subjects? Can you provide details about your process for gaining this understanding and how you will apply it to your data curation project?

- How do you intend to prioritize safeguarding data subjects' rights as stipulated by the data protection laws in their respective countries? What steps will you take to ensure that the creation and utilization of the dataset strictly adhere to the relevant data protection regulations? Can you provide specifics about the measures you will put in place to achieve this?

- What mechanisms do you intend to implement to ensure the adaptability of your dataset management strategy to changing legislative requirements? Can you provide details about how you will monitor and accommodate legislative changes in your dataset management approach? Can you provide insights into how you will strike a balance between maintaining compliance and effective dataset management in dynamic legal environments?

**Privacy-Sensitive Image Regions and Metadata**

- How do you plan to implement measures that effectively safeguard against re-identification risks, encompassing singling out, linkability, and inference, within your anonymization approach?

- Can you elaborate on your strategy for redacting all image regions that could inadvertently disclose privacy-related information? How do you intend to comprehensively identify and address these regions?

- Can you elaborate on your strategy for the removal of elements such as body parts, clothing, and accessories for nonconsenting subjects to enhance privacy protection? Can you provide more details about the considerations and methods involved in this process?

- Can you elaborate on your strategy for the removal of text (possibly excluding copyright owner information) from the dataset's images to enhance privacy protection? Can you provide more details about the considerations and methods involved in this process?

- Can you explain your plan for empirically validating the chosen anonymization methods? How will you assess the methods' effectiveness in mitigating re-identification risks while preserving the utility of the data?

- Can you provide details about how human annotators will be engaged in the creation and verification of privacy leaking image region proposals for anonymization purposes? How will you ensure accuracy and consistency in this process?

- Can you provide details about how you intend to align region proposals predicted by algorithms with human judgment, addressing any potential failures or biases? Can you describe your strategy for maintaining a sensitive approach to these factors?

- What steps will you take to address jurisdiction-specific requirements that might necessitate human-generated proposals for biometric identifiers in order to comply with legal and regulatory standards?

- Can you elaborate on the measures you will take to prevent image metadata from inadvertently revealing unauthorized identifying information? How will you ensure that metadata remains privacy-conscious?

- How will you identify specific metadata elements that you intend to retain to ensure a comprehensive understanding during the evaluation process? Can you provide examples of the types of metadata you plan to retain for this purpose?

- How do you plan to replace or remove sensitive information within metadata while retaining its usefulness for fairness and robustness analyses? Can you provide insights into your approach for striking a balance in this regard?

## A.3 Diversity

The questions in this section revolve around obtaining accurate image annotations related to identity, phenotype, environmental factors, and instruments, while upholding inclusivity, sensitivity, and privacy. Additionally, the questions attempt to elicit strategies for documenting identity, ensuring fair compensation, and effective (anonymous) communication.

**Self-Reported Annotations**

- How do you plan to acquire annotations for images directly from the data subjects, leveraging their self-awareness and contextual knowledge to enhance the accuracy and quality of annotations? Can you elaborate on the methods and strategies you intend to use for this purpose?

- Can you elaborate on your strategy for addressing biases and ensuring careful handling when inferring labels about individuals? Can you provide reasoning as to why labels about individuals will be inferred as opposed to being self-identified? How will you actively mitigate potential biases that may arise during the labeling process?

- How do you intend to consider the implications of inferred labels, for example, in relation to data access request rights?

**Versatile and Inclusive Response Options**

- How do you plan to enhance the accuracy and nuance of identity information collection by providing respondents with both closed-ended and open-ended response choices? Can you elaborate on your strategy for using open-ended responses to gather more detailed and comprehensive data?

- How do you intend to ensure inclusivity and prevent any potential implications of exclusion in the response choices you offer?

- Can you elaborate on your preparedness to manage the coding and analysis effort required for processing open-ended responses? What effective strategies do you plan to implement for managing and analyzing the data collected from open-ended questions? How will you handle the potential complexities and variations that can arise from these responses, ensuring that the insights and information derived can be accurately captured and utilized?

**Dynamic Nature of Identity**

- How do you plan to collect self-identified information on a per-image basis, accounting for the fact that identity is intrinsically contextual and temporal? Can you elaborate on your strategy for capturing nonstatic aspects of identity?

- Can you elaborate on your strategy for enabling data subjects to freely choose multiple identity categories without imposing any limitations? How will you ensure that subjects have the flexibility to express their identity in a comprehensive and unrestrictive manner?

- How do you intend to address potential requests for per-image updates to self-identified information provided by subjects over time, respecting their autonomy? What factors have you considered in relation to the potential effects of permitting updates?

**Demographic Information**

- How do you plan to collect precise biological age in years from data subjects to ensure an accurate representation of their age?

- Can you elaborate on your approach to gathering pronoun information from data subjects to enhance gender inclusivity and mitigate the risk of misgendering? How will you ensure that respondents feel comfortable providing this information?

- Can you explain your strategy for gathering consistent ancestry information from data subjects? How will you approach the collection of this information in a sensitive and inclusive manner?

- How do you intend to offer the option for data subjects not to disclose their sensitive attributes if they choose not to? Can you provide more details about how you will handle the sensitivity and privacy of these attributes?

**Sensitive Attributes in Aggregate**

- How do you plan to collect voluntarily disclosed sensitive attributes such as disability and pregnancy status? Can you elaborate on your approach to respecting the willingness of data subjects to provide these details?

- Can you provide insight into your strategy for reporting sensitive attributes, such as disability and pregnancy status, in aggregate data while safeguarding subjects' safety and privacy? How do you intend to ensure that individual identities are protected?

- Can you elaborate on your approach to relying on credible and appropriate sources for the categorization and definitions of sensitive attributes like disability or difficulty? How will you account for the potential variations in these definitions based on cultural, identity, and historical contexts?

**Phenotypic and Neutral Performative Features**

- How do you plan to collect phenotypic attributes, encompassing characteristics such as skin color, eye color, hair type, hair color, height, and weight? Can you provide insights into your strategy for obtaining these attributes in a sensitive and comprehensive manner?

- Can you elaborate on your approach to collecting a diverse range of neutral performative features, including aspects such as facial hair, hairstyle, cosmetics, clothing, and accessories? How do you intend to ensure inclusivity and accuracy in capturing these features?

**Environment and Instrument Details**

- How do you plan to gather data on environment-related factors, which encompass details such as image capture time, season, weather, ambient lighting, scene, geography, camera position, and camera distance? Can you provide insights into your strategy for capturing these factors accurately and comprehensively?

- Can you elaborate on your approach to collecting instrument-related factors concerning the imaging devices used, including aspects such as lens, sensor, stabilization, flash usage, and post-processing software? How do you intend to ensure accuracy in capturing these details?

- How do you plan to obtain environment- and instrument-related information? Can you provide more details about the methods you will use, such as self-reporting, annotator estimation, and sourcing information from Exif metadata? How will you leverage contextual knowledge from image subjects to enhance data quality?

- Can you explain your approach to handling information such as precise geolocation and user-added details in Exif metadata that might contain personally identifying information? How will you ensure compliance with copyright regulations (if applicable) while maintaining privacy and adhering to ethical considerations?

## Annotators as Contributors

- How do you plan to document the identities of data annotators, including capturing demographic details such as pronouns, age, and ancestry? Can you provide insights into your strategy for gathering and preserving this information while respecting privacy and ensuring transparency?

- Can you elaborate on your approach to highlighting the contributions of annotators beyond data labeling in the dataset documentation after the curation process? How do you intend to accurately represent the multifaceted roles and contributions of annotators?

- How do you plan to report the demographic information of annotators to analyze potential sources of bias in dataset annotations? Can you provide more details about your proposed approach for conducting this analysis while ensuring privacy and ethical considerations?

## Fair Treatment and Compensation

- How do you plan to ensure that all contributors receive compensation that exceeds the minimum hourly wage of their respective country or jurisdiction of residence? Can you provide insights into your compensation strategy to ensure fair and ethical remuneration?

- Can you elaborate on your approach to exploring alternative payment models, such as compensation based on the average hourly wage? How do you intend to determine a compensation structure that is both fair and reflective of contributors' efforts?

- How will you establish direct communication channels between dataset creators and contributors? Can you provide more details about the methods you intend to implement for effective and transparent communication?

- What communication methods do you plan to explore that maintain the anonymity of contributors? Can you provide insights into your approach to balancing communication and privacy needs, such as using anonymous feedback forms?

- Can you provide information about your strategy for developing clear and accessible plain language guides to facilitate various tasks, such as image submission and data annotation? How do you plan to ensure that these guides effectively assist contributors?

- How do you intend to ensure that contributors from diverse backgrounds can easily understand and follow any instructions provided? Can you elaborate on your approach to promoting inclusivity and accessibility in your communication and guidelines?

- Can you provide details about how you plan to subject your recruitment and compensation procedures to ethics review? What steps will you take to ensure that your procedures align with ethical considerations and best practices?

# B    Literature Review

Through a thematic search strategy, we identified relevant research studies and datasets, revealing deficiencies in current image data curation practices or proposing potential solutions. By utilizing Semantic Scholar and Google Scholar, we curated relevant papers covering a wide spectrum of themes, including:

- HCAI
- Human-subjects research
- HCCV datasets
- Dataset curation
- Ethical frameworks and considerations
- Data and model documentation
- Legal and regulatory considerations
- Privacy and data protection
- Consent
- Fairness
- Auditing and verification

- Guidelines and best practices
- Values in design
- Diversity and inclusion
- Representation
- Robustness and reliability
- Benchmarking and evaluation
- Bias detection and mitigation
- Critical AI
- Social implications
- Responsible AI

The themes were chosen based on our expertise and experience in designing CV datasets, training models, and developing ethical guidelines. To ensure a focused approach, we manually selected papers aligned with the scope of our study based on the relevance of a paper's title and abstract. This informed our initial categorization scheme, shown in Table 1, detailing key ethical considerations related to HCCV.

Initially broad, we further refined the categories to address the most prominent ethical issues pertaining to HCCV dataset curation, particularly for fairness and robustness evaluations. Consent and privacy categories were combined due to their interrelated nature and the influence of shared legal frameworks.

Table 1: Categorization scheme for ethical considerations in HCCV research

| Category | Explanation |
| --- | --- |
| Purpose | The study discusses the underlying objectives and motivations for HCCV datasets. |
| Acquisition | The study discusses ethical considerations related to the acquisition, collection, and labeling of image data, including recruitment and compensation for contributors. |
| Consent | The study discusses consent and the responsible use of personal information. |
| Privacy | The study discusses privacy issues related to HCCV datasets or public data. |
| Ownership | The study discusses legal and ethical aspects of intellectual property rights in the context of HCCV datasets or public data. |
| Diversity | The study discusses factors concerning diversity, inclusion, and fair representation within HCCV datasets. This encompasses matters such as identifying and addressing biases, ensuring fairness, and mitigating discrimination. |
| Maintenance | The study discusses maintenance strategies for ensuring the integrity of HCCV datasets, including security measures. |

Additionally, we integrated acquisition-related considerations into the categories of diversity, consent and privacy, as well as purpose, recognizing their interconnectedness in ethical image data collection, labeling, and usage. Maintenance-related matters were intentionally excluded from our scope, as these primarily pertain to post-dataset creation activities, while technical and organizational security measures are typically covered through consent forms. Ownership concerns, often intertwined with privacy issues, were incorporated into the consent and privacy category.

To establish a comprehensive view, we expanded our corpus as necessary. This encompassed examining cited works within our initial corpus, studies referencing our primary sources, and additional contributions by authors from our initial corpus. Our review was supplemented by incorporating publicly available resources from reputable sources, such as government bodies, private institutions, and reliable news organizations. In total, our analysis covered 500 research studies and online resources.

