# OpenReview forum: "Ethical Considerations for Responsible Data Curation"
_NeurIPS.cc/2023/Track/Datasets_and_Benchmarks — NeurIPS 2023 Datasets and Benchmarks Oral_

### Official Review · Reviewer_qLN4 · 2023-07-19
**Review of #258: Principlism Guided Responsible Data Curation**

**Rating:** 7
**Confidence:** 4

**Strengths:**

The key strength of the paper lies in the broad depth of literature surveyed and synthesised - 331 sources [covered in] 20 pages. The depth of work needs to be acknowledge and this reviewer commends the author team for this.

In addition, the authors have narrowed down specific issues pertaining to HCCV datasets, which is an attempt at addressing the unique issues not found in others such as LLM training corpora (e.g., "privacy leaking image regions and metadata", p.5). In addition, the recommendations provided are guided by legal considerations (the ubiquitous GDPR, say) in tandem with academic studies.

Further, the authors are aware of the existing sets of broad strategies in different working groups/committees/government agencies and attempt to tie these into the suggestions raised in their "Practical recommendations" sections.

* EDIT: "covered in 20 pages"

**Additional Feedback:**

This reviewer would encourage the authors to instead pitch the submission as a literature review and position paper -- more importantly, with the aims of establishing a new *framework* on data curation in HCCV - which will potentially be more topical, contextual, and very likely to succeed (in this reviewer's opinion).

This includes, e.g. potential templates or actionable prompts in the same direction as Mitchell et al's Model Cards or Gebru et al's Datasheets initiatives which are compact, self-contained, and [more importantly] easy-to-get-uptake from the researcher community -- their position papers create a framework, guided by extant research, and are easily applicable.

A major issue as mentioned above: the main claims of 'principlism-guided' need to be reconsidered, or removed entirely. The Beauchamp & Childress concepts are not engaged with at depth; and the claim of "principlism-guided" needs to be tempered.

**Clarity:**

In the main, it is quite well written with no issues with language nor presentation. The authors are commended for this.

**Correctness:**

In terms of *Correctness* (expanding from *Opportunities For Improvement*), I'd suggest that if the Beauchamp & Childress 4 Principles are to be engaged with, it should be more than just brief mentions of the four principles. A search through the document for each of the four principles yields at most 5 mentions. The methodology of "To align our expertise with the ethical framework, we collaboratively discussed principlism’s four pillars, considering each author’s background" might not suffice here -- a reviewer would expect, say, for the sake of argument:
> "Consent revocation" (p.4) aligns with the principle of "autonomy" because the data subject needs to be able to have "decision-making rights" in the spirit of Kantian philosophy (i.e., using others not merely as a means in achieving, say, data saturation -- cf. 2nd Categorical Imperative) -- citing Beauchamp & Childress (2001).

Further, a misconception that might need to be addressed is that the Beauchamp & Childress 4 Principles are meant to be used as an easy set of heuristics in decision-making, and not merely used in setting the scene for a literature review -- this has to be again *emphasised in this contribution if it were to maximise the usefulness of these principles*. Again, pick any recommendation such as "Obtain self-reported annotations" -- autonomy plays a role here (one would argue being the primary principle at work); and so is "nonmaleficence" and "justice" as the fact that misclassification is a form of *epistemic hermeneutical injustice* cf Miranda Fricker (2007). Such argumentation will help solidify the benefit of these principles and guide the reader to *why* recommendations provided are grounded in Beauchamp & Childress.

**Documentation:**

[The submission is pitched as a set of recommendations towards improving the ethical landscape on research, and hence this section does not apply.]

**Ethics:**

[The submission is pitched as a set of recommendations towards improving the ethical landscape on research, and hence this section does not apply.]

**Limitations:**

[The submission is pitched as a set of recommendations towards improving the ethical landscape on research, and hence this section does not apply.]

**Opportunities For Improvement:**

The paper has some opportunities for improvement. For starters, the evidence for Principlism as an *advantage* is lacking - beyond several mentions at the start and towards the end, the use of these Beauchamp & Childress principles are not covered in detail nor justified (see *Correctness* below, in particular 1. not considering other frameworks and justifying the choice; and 2. lack of contextualisation for each issue with the Beauchamp & Childress 4 principles).

The authors also claim (I quote) "[c]urrent remedies address issues post hoc, lack persuasive justification for adoption, or fail to provide proper contextualization for appropriate application" -- while this paper makes a solid attempt at the middle criterion, and perhaps the latter criterion, many valid points raised in this paper *can be used in a post-hoc fashion* as well.

Lastly, there is a risk of 'missing the forest for the trees', in that (1) the submission has a prior assumption that "bypassing Institutional Review Board supervision" is the *norm* rather than the exception (p.3) -- having familiarity with data governance and IRB processes, this reviewer begs to differ. As a result, this submission seems to be creating an *additional* onus on potential experimenters to go through in addition to IRBs, which leads to (2) potential lack of uptake -- when compared against, e.g., Mitchell et al's model cards which are designed to be compact, thought-provoking, and most importantly complements existing processes -- which is a buy-in for adoption.

**Relation To Prior Work:**

As discussed above: this work differs from existing ones which are "post hoc, lack persuasive justification for adoption, or fail to provide proper contextualization for appropriate application".

**Summary And Contributions:**

To summarize, this submission is about an ante-hoc "reflective perspective" on ethical considerations for curation of datasets (specifically HCCV - Human-centric Computer Vision). The authors claim that this differs from existing ones which are "post hoc, lack persuasive justification for adoption, or fail to provide proper contextualization for appropriate application".

In the contribution, the authors review a significant body of literature and highlight broad categories of issues with the three broad themes of Purpose / Consent and Privacy / Diversity. Each category has ethical issues and considerations identified from extant literature; and suggested ameliorative recommendations are provided for each issue/consideration.

---

> ### Author Response · Authors · 2023-08-17
> **Author Response**
>
> Thank you for your insightful review. We particularly appreciate your recognition of our paper’s extensive literature review, clarity and presentation, as well as its distinctiveness in relation to prior work. In addition, thank you for raising areas for improvement.
>
>
> ---
>
> Please see our responses addressing Opportunities for *Improvement*, *Correctness*, and *Additional Feedback* below.
>
>
> > there is a risk of 'missing the forest for the trees', in that (1) the submission has a prior assumption that "bypassing Institutional Review Board supervision" is the norm rather than the exception (p.3) -- having familiarity with data governance and IRB processes, this reviewer begs to differ.
> >
> > As a result, this submission seems to be creating an additional onus on potential experimenters to go through in addition to IRBs, which leads to (2) potential lack of uptake -- when compared against, e.g., Mitchell et al's model cards which are designed to be compact, thought-provoking, and most importantly complements existing processes -- which is a buy-in for adoption.”
>
> Addressing (1), we have expanded upon this point in the revised paper (see Lines 57-74 in the *Development Section*) and below:
>
> HCCV should adhere to the most stringent ethical standards to address privacy and bias concerns. As stated in the NeurIPS Code of Ethics [1], it is essential to abide by established institutional research protocols, ensuring the safeguarding of human subjects. These protocols, initially designed for biomedical research, have, however, been met with confusion, resulting in inconsistencies when applied in the context of data-centric research [217]. For example, HCCV research often amasses millions of public images without obtaining informed consent or participation, disregarding serious privacy and ethical concerns [35, 132, 244, 300, 3]. This exemption from research ethics regulation is grounded in the limited definition of human-subjects research, which categorizes extant, publicly available data as minimal risk [255, 217]. Thus, numerous ethically-dubious HCCV datasets would not fall under Institutional Review Board (IRB) oversight [246]. What’s more, the NeurIPS Code of Ethics only mandates following existing protocols when research involves direct *interactions* between human participants and researchers or technical systems. Even when research is subjected to supervision, IRBs are restricted from considering broader societal consequences beyond the immediate study context [216]. Compounding matters, CV-centric conferences are still to adopt ethics review practices [305]. In combination, this is problematic due to the potential for predictive privacy harms when seemingly non-identifiable data is combined [69, 217, 35] or when data is used for harmful downstream applications such as predicting sexual orientation [342, 191], crime propensity [356, 363], or emotion [223, 10].
>
> Addressing (2), we have now created an Appendix. In *Appendix A*, similar to existing AI ethics checklists [204, 224, 268, 355, 200, 89], we have created a checklist which translates each of our considerations and recommendations into action items for researchers and practitioners. Presented as a series of questions, mirroring the structure of our paper, these items are designed to stimulate discussions among data collection teams. The checklist is intended to be engaged with as a preliminary exercise before beginning data collection, promoting informed decision-making and minimizing risks. The checklist does not guarantee ethical compliance; rather, it functions as a catalyst for discussion and reflection.
>
> Our intention is that this addition aids data collection teams in transitioning, or at the very least, prompts contemplation about adopting more ethical practices.

---

> ### Author Response · Authors · 2023-08-17
> **[Continued] Author Response**
>
> > A major issue as mentioned above: the main claims of 'principlism-guided' need to be reconsidered, or removed entirely. The Beauchamp & Childress concepts are not engaged with at depth; and the claim of "principlism-guided" needs to be tempered.
>
>
> * We have updated the title of our paper from *Principlism Guided Responsible Data Curation* to *Considerations for Responsible Data Curation*
> * The paper is entirely focused on our extensive literature review of current practices, guidelines, dataset withdrawals, and audits, which inform our considerations and recommendations. As mentioned by the reviewer, principlism was not engaged with at depth, and as such its removal is inconsequential to our comprehensive literature review and recommendations that are motivated and explained in detail.
> * However, we still maintain within our *Development Process* section that "we employed the same principles underpinning established guidelines [24, 329] for protecting human subjects in research to *identify* ethical issues in HCCV dataset design". The use of these principles to identify ethical issues is well motivated by the limitations we raised above (Lines 57-94 in the revised paper), regarding IRBs, the NeurIPS Code of Ethics, and ethics review processes.
>
>
>
> > The authors also claim (I quote) "[c]urrent remedies address issues post hoc, lack persuasive justification for adoption, or fail to provide proper contextualization for appropriate application" -- while this paper makes a solid attempt at the middle criterion, and perhaps the latter criterion, many valid points raised in this paper can be used in a post-hoc fashion as well.
>
> We agree that several of our recommendations can be applied in a post-hoc fashion. However, our point was that our considerations and recommendations should be mediated over prior to data collection.
>
> We have now further clarified this in the revised paper (see Lines 51-53). That is, "While several of our recommendations can also be applied retroactively such measures cannot undo incurred harm, e.g., resulting from inappropriate uses, privacy violations, and unfair representation [128]."
>
> For example, offering bonuses to data annotators to ensure they earn at least minimum wage is a positive step. Nonetheless, this action does not erase any harm that may have occurred before implementing it. That is, while the annotators' compensation was below minimum wage.
>
> Similarly, consider the situation where Exif metadata is absent from images due to, e.g., post-processing or device limitations. This absence can make it difficult to gauge the influence of instrument-related factors. Unfortunately, there is no way to address this post-data collection, which could potentially limit the thoroughness of fairness and robustness assessments.
>
> To address these issues, it is important to proactively plan before collecting data. For example, this may involve considering factors such as permissible devices (i.e., capable of recording Exif) to ensure that potential challenges are anticipated and mitigated from the outset.
>
>
> ---
>
> Let us know if these additions are satisfactory. Thank you again for taking the time to review our paper and the opportunity to improve the paper.

---

> > ### Comment · Reviewer_qLN4 · 2023-08-18
> > **Reply to rejoinder**
> >
> > Thank you, authors, for the comprehensive review.
> > I am of the opinion that *Appendix A is a strong addition which is of utility*, and that *removing the "principlism-guided" claim* (which was one of my major concerns re Beauchamp&Childress et al) have ameliorated the substantial concerns I've had.
> > The score has been revised accordingly, for transparency, from "3" to "7".

---

### Official Review · Reviewer_c5ch · 2023-07-19
**A thorough overview of current HCCV data curation practises and domain-specific recommendations.**

**Rating:** 9
**Confidence:** 3
**Correctness:** It seems that the claims are adequate…
**Clarity:** I found the paper clear, concise, and…

**Strengths:**

- This work has been written by authors of diverse backgrounds, and research domains, which makes a crucial strong point for this kind of work.
- The paper is well-written and nicely presented by the authors.
- The work focuses on open research challenges and issues revolving around the topic.

**Additional Feedback:**

Good work overall. I really liked that the authors compose a diverse research group of backgrounds and research domains that are of high importance for this kind of contribution.

**Documentation:**

Not applicable to this kind of contribution.

**Ethics:**

I have not found any ethical concerns to address.

**Limitations:**

I would really like to see how they gathered the respective literature and what they might didn't explored, also mentioning it as future work.

Also, I haven't seen any dedicated limitations discussion in the paper.

**Opportunities For Improvement:**

I would really like to see some case studies or a kind of evaluation of these recommendations in order to demonstrate if they are practically feasible to be applied.

**Relation To Prior Work:**

The authors adequately provided and compared points related to previous work in specific discussions in the paper.

**Summary And Contributions:**

The authors provide a thorough overview of current human-centric computer vision (HCCV) data curation practices. They primarily focus on proposing proactive, domain-specific recommendations for curating HCCV datasets, addressing privacy and bias, guided by current practices, guidelines, and the ethical framework of principlism.

---

> ### Author Response · Authors · 2023-08-17
> **Author Response**
>
> Thank you for the very positive feedback, we are pleased that you found our paper to be thorough and well-presented. Also we appreciate your positive recognition of our efforts to assemble a diverse research group with backgrounds and research domains relevant to this work.
>
> ---
>
> Please see our responses addressing *Limitations* below.
>
> > I would really like to see how they gathered the respective literature and what they might didn't explored, also mentioning it as future work.
>
> We have added additional details to the *Development Section* in the paper. In addition, we have created an Appendix. In *Appendix B*, we present details on our literature review, which we summarize below.
> * Through a thematic search strategy, we identified relevant research studies and datasets, revealing deficiencies in current image data curation practices or proposing potential solutions. By utilizing Semantic Scholar and Google Scholar, we curated relevant papers covering a wide spectrum of themes, including HCAI, human-subjects research, HCCV datasets, ethical considerations, legal aspects, privacy, fairness, diversity, representation, critical AI, and more.
> * The themes were chosen based on our expertise and experience in designing CV datasets, training models, and developing ethical guidelines.
> * To ensure a focused approach, we manually selected papers aligned with the scope of our study based on the relevance of a paper’s title and abstract. We present our initial categorization scheme in *Appendix B* Table 1, detailing key ethical considerations related to HCCV.
> * Initially broad, we further refined the categories to address the most prominent ethical issues pertaining to HCCV dataset curation, particularly for fairness and robustness evaluations. Consent and privacy categories were combined due to their interrelated nature and the influence of shared legal frameworks. Additionally, we integrated acquisition-related considerations into the categories of diversity, consent and privacy, as well as purpose, recognizing their interconnectedness in ethical image data collection, labeling, and usage. Maintenance-related matters were intentionally excluded from our scope, as these primarily pertain to post-dataset creation activities, while technical and organizational security measures are typically covered through consent forms. Ownership concerns, often intertwined with privacy issues, were incorporated into the consent and privacy category.
> * To establish a comprehensive view, we expanded our corpus as necessary. This encompassed examining cited works within our initial corpus, studies referencing our primary sources, and additional contributions by authors from our initial corpus. Our review was supplemented by incorporating publicly available resources from reputable sources, such as government bodies, private institutions, and reliable news organizations.
> * In total, our analysis covered 500 research studies and online resources.

---

> ### Author Response · Authors · 2023-08-17
> **[Continued] Author Response**
>
> > Also, I haven't seen any dedicated limitations discussion in the paper.
>
> We have replaced our *Conclusion* section with a *Discussion* section, where we now acknowledge the following limitations:
> * Implementing our proposed recommendations could face resistance due to established norms [217], inertia [33], diffusion of responsibility [150], and liability concerns [15].
> * Seeking consent from all depicted individuals could be logistically challenging, requiring investment in consent management systems and personnel. Particularly for smaller organizations and academic research groups, these limitations could present considerable hurdles.
> * Scaling our recommendations to large datasets, such as democratizing foundation model-sized training datasets [118, 287, 288, 103], presents an economic challenge. For example, the GeoDE dataset of ~67k crowdsourced object-centric images [261], without personally identifiable information, incurred a cost of $1.08 per image.
>
>
> In addition, within the *Discussion* section, we provide some recommendations and thoughts on how to address the above limitations:
>
> * Platforms such as NeurIPS could adopt a model similar to the registered reports format, embraced by over 300 journals [50, 246]. This entails pre-acceptance of dataset proposals before curation, alleviating financial uncertainties associated with implementing more ethical practices. (As detailed in Section 3.2, the UK's 2021 Research Excellence Framework has acknowledged the significance of registered reports. This recognition suggests that authors who opt for publishing registered reports might potentially secure augmented funding for their respective institutions [50].)
> * Smaller organizations and academic research groups, potentially alongside larger institutions, could form data consortia [165, 120], which can help to address operational challenges by pooling resources and knowledge. A successful, prominent example is the Ego4D consortium [120].
> * While our recommendations may not seamlessly scale to the curation of fairness-aware, billion-sized image datasets, it is worth considering that “solutions which resist scale thinking are necessary to undo the social structures which lie at the heart of social inequality” [129]. Large-scale, nonconsensual datasets, driven by scale thinking, have included harmful and distressing content, including rape [36, 35], racist stereotypes [130], depictions of children [127], and derogatory taxonomies [182, 34, 128, 68]. The inclusion of such materials may further generate legal concerns [2]. We contend that these issues can be mitigated through the implementation of our recommendations.
> * Balancing resources between model development and data curation is a value-laden choice shaped by “social, political, and ethical values” [38]. While organizations readily invest significantly in model training [226, 334], compensation for data contributors often appears neglected [336, 335], disregarding that “most data represent or impact people” [378]. Remedial actions could be envisioned to bridge the gap between models developed with ethically curated data and those benefiting from expansive, nonconsensual crawled data. Reallocating research funds away from dominant data-hungry methods [38] would help to strike a balance between technological advancements and ethical imperatives. That being said, the granularity and comprehensiveness of our diversity recommendations could be adapted beyond evaluation contexts, particularly when employing “fairness without demographics” [209, 188, 133, 49] training approaches, reducing financial costs.
>
>
> ---
>
> Let us know if these additions are satisfactory. Thank you for taking the time to review our paper and providing such positive feedback.

---

> > ### Comment · Reviewer_c5ch · 2023-08-29
> >
> > I would like to thank the authors for taking the time and effort to address my comments on their current work. As my main concern was about the limitations revolving this research, I believe that the authors have adequately addressed them by appending the respective changes in the paper. As a result, I’m raising my score from 8 to 9.
> >
> > I’m looking forward to see this work getting published and the upcoming work that would leverage those considerations.

---

### Official Review · Reviewer_1b4S · 2023-07-21
**Principlism Guided Responsible Data Curation**

**Rating:** 9
**Confidence:** 5
**Correctness:** As far as is possible, yes.
**Clarity:** Very well written.

**Strengths:**

The paper is an excellent summary of the existing literature that points to the need for diversity, agency, sensitivity, and fairness in HCCV datasets. It makes contributions regarding the confused taxonomies of attributes like sex and gender, race and ethnicity, biased annotator influences, and ethical considerations such as data collection without consent, noninclusive labelling practices, and under-compensation of remote workers. The greatest strength is its specific practical recommendations, demonstrating a deep understanding of the need for diversity and inclusivity in dataset creation. The authors make a strong case for adopting self-reported data and open-ended response options. Although the authors note the mutability and multiplicity of identity, it raises the question of whether the kind of very granular information they suggest gathering allows for this kind of flexibility. The emphasis is put on data subjects to opt out or modify their self assessments.

A standout aspect of these recommendations is their focus on fair and equitable treatment of contributors. The authors argue for a recognition system that goes beyond just compensation, and includes clear communication channels.




**Additional Feedback:**

This is an excellent paper that contributes to the work on responsible data curation.

**Documentation:**

NA

**Ethics:**

No.

**Limitations:**

Again, this paper would benefit from more detail on the scale and specificity of potential harms, and being more direct about examples of dehumanizing and discriminatory impacts of datasets. It should be more detailed about the kinds of harms already discovered (e.g Crawford and Paglen 2019; Prabhu and Birhane 2021). Otherwise practitioners may see the excellent recommendations as more 'ideal if possible', rather than necessary to avoid serious consequences for data subjects and model performance.

**Opportunities For Improvement:**

The one thing I would suggest the authors reflect on further is connecting their interpretation of principlism with its origins in bio-ethics, and particularly the dark history of why human subjects processes were instituted in the first place. That is, the paper could state more clearly the human rights failures of HCCV so far, and who has been most harmed. How should that increase the responsibility of practitioners, and in particular, why shouldn't they commit to doing human subjects review? The paper currently states that as 'ideal' but not necessary, and this could benefit from being strengthened, particularly given the kinds of harms that are possible.

**Relation To Prior Work:**

The paper is excellent at reviewing prior literature.

**Summary And Contributions:**

This paper addresses the state of human-centric computer vision (HCCV) datasets, and observes how they prioritize dataset size and utility while often ignoring privacy and bias concerns, leading to unfair models and dataset retractions. These practices heavily rely on nonconsensual web scraping, which may result in models that lack necessary metadata for fairness evaluations. The authors propose an 'ante hoc' approach that integrates ethical considerations from the start, focused on purpose, consent, privacy, and diversity, guided by the ethical framework of principlism (autonomy, beneficence, non-maleficence, justice).

---

> ### Author Response · Authors · 2023-08-17
> **Author Response**
>
> Thank you for the very positive feedback and appreciation that our work represents an excellent contribution to the field of responsible data curation.
>
> ---
>
> Please see our responses addressing *Opportunities for Improvement* and *Limitations* below.
>
> > Although the authors note the mutability and multiplicity of identity, it raises the question of whether the kind of very granular information they suggest gathering allows for this kind of flexibility. The emphasis is put on data subjects to opt out or modify their self assessments.
>
> We have clarified this in the revision (see Lines 383-386). That is, we have acknowledged the potential burden of self-identification on fluid and dynamic identities. Nonetheless, acknowledging the mutability and multiplicity of identity is necessary to prevent oversimplification and marginalization.
>
> Moreover, we have acknowledged possible requests from data subjects for metadata updates. We assert that an image only captures a single moment in time and space. Evolving identity, therefore, may not require metadata updates. Nevertheless, we have augmented our recommendation by additionally recommending the provision of mechanisms that empower data subjects to update metadata for previously submitted images.
>
> > ... the paper could state more clearly the human rights failures of HCCV so far, and who has been most harmed. How should that increase the responsibility of practitioners, and in particular, why shouldn't they commit to doing human subjects review?
> > ... this paper would benefit from more detail on the scale and specificity of potential harms, and being more direct about examples of dehumanizing and discriminatory impacts of datasets. It should be more detailed about the kinds of harms already discovered (e.g Crawford and Paglen 2019; Prabhu and Birhane 2021).
>
>
> * In the *Development Process* section, we have added a few examples of harmful applications (see Lines 73-74).
> * In the new *Discussion* section, we have added examples of harmful and distressing content in large-scale datasets.
> * We have included an Appendix as well. In *Appendix C*, we showcase a range of notable instances displaying discriminatory outcomes within HCCV, spanning from 2011 to 2022. Furthermore, we underscore the obligation that researchers and practitioners hold in addressing broader societal repercussions. This responsibility is particularly pertinent due to the limited scope of IRBs, which primarily focus on immediate research study contexts rather than broader societal impacts. Due to formatting constraints in the NeurIPS D&Bs Track's "submitted to" option, which differs from the "camera ready" option, we were unable to incorporate the contents of *Appendix C* into the main text. Should our paper be accepted, our intention is to make this inclusion.
>
>
>
>
> ---
>
> Let us know if these additions are satisfactory. Thank you for taking the time to review our paper and providing such positive feedback.

---

### Official Review · Reviewer_so9D · 2023-07-22
**Review for "Principlism Guided Responsible Data Curation"**

**Rating:** 9
**Confidence:** 4

**Strengths:**

This paper is exceptionally thorough and well referenced. I found it easy to read and very interesting. The authors have distilled the existing literature and focused their synthesis into practical recommendations that can be put in place by human centric computer vision researchers. I expect that readers working in this field will bookmark this paper and return to it often.

**Additional Feedback:**

I look forward to a future where these recommendations are implemented!! Thank you for your work.

**Clarity:**

Very well written, very clear. I found it easy to read and understand and I expect many others will come back to it regularly for the practical recommendations, the overarching ethical considerations, and the references.

**Correctness:**

This paper is exceptionally well referenced and represents an accurate assessment of ethical considerations around responsible data curation.

**Documentation:**

Not applicable.

**Ethics:**

No ethical concerns.

**Limitations:**

I don't see any negative social impact of this work. This paper is very strongly encouraging the reader to work more responsibly with data for human centric computer vision. They are working to undo some of the existing negative social impact.

The scope of the paper is well explained and I don't see any limitations that have not been covered.


**Opportunities For Improvement:**

My only quibble is that the "practical recommendations" are - from some perspectives - not very practical at all. I would have loved to see an acknowledgement of what could NOT be done in HCCV if these recommendations were put in place. The cost of collecting data would be higher I expect. But how much higher? How should that be estimated if we can't use existing datasets (for clearly explained ethical reasons)? How much data is needed? Does the enhanced curation mean models can be run on less? (I expect so) But how much less? Who would know.

I think this paper would be additionally enhanced and the recommendations more likely to be adopted if the authors could help HCCV researchers to make the transition from current ways of working to these much more ethical, but person time intensive, methods.

**Relation To Prior Work:**

This work synthesises the existing literature and presents actions specific to human centric computer vision. The work builds on but represents a distinct and important contribution to the field.

**Summary And Contributions:**

This paper presents a detailed list of decisions and actions for data curation in human-centric computer vision. There are ethical considerations and practical recommendations for 1) considering the purpose of the dataset, 2) respecting individuals' consent and privacy, and 3) understanding the diversity of the people represented in the dataset.

---

> ### Author Response · Authors · 2023-08-17
> **Author Response**
>
> Thank you for very positive feedback and your kind words. We are particularly pleased that you envision our paper as a significant bookmark that readers in this field will regularly revisit.
>
> ---
>
> Please see our responses addressing *Opportunities for Improvement* below.
>
> > I would have loved to see an acknowledgement of what could NOT be done in HCCV if these recommendations were put in place.
>
> We have replaced our *Conclusion* section with a *Discussion* section, where we now acknowledge the following limitations:
> * Implementing our proposed recommendations could face resistance due to established norms [217], inertia [33], diffusion of responsibility [150], and liability concerns [15].
> * Seeking consent from all depicted individuals could be logistically challenging, requiring investment in consent management systems and personnel. Particularly for smaller organizations and academic research groups, these limitations could present considerable hurdles.
> * Scaling our recommendations to large datasets, such as democratizing foundation model-sized training datasets [118, 287, 288, 103], presents an economic challenge. For example, the GeoDE dataset of ~67k crowdsourced object-centric images [261], without personally identifiable information, incurred a cost of $1.08 per image.
>
>
> In addition, within the *Discussion* section, we provide some recommendations and thoughts on how to address the above limitations:
>
> * Platforms such as NeurIPS could adopt a model similar to the registered reports format, embraced by over 300 journals [50, 246]. This entails pre-acceptance of dataset proposals before curation, alleviating financial uncertainties associated with implementing more ethical practices. (As detailed in Section 3.2, the UK's 2021 Research Excellence Framework has acknowledged the significance of registered reports. This recognition suggests that authors who opt for publishing registered reports might potentially secure augmented funding for their respective institutions [50].)
> * Smaller organizations and academic research groups, potentially alongside larger institutions, could form data consortia [165, 120], which can help to address operational challenges by pooling resources and knowledge. A successful, prominent example is the Ego4D consortium [120].
> * While our recommendations may not seamlessly scale to the curation of fairness-aware, billion-sized image datasets, it is worth considering that “solutions which resist scale thinking are necessary to undo the social structures which lie at the heart of social inequality” [129]. Large-scale, nonconsensual datasets, driven by scale thinking, have included harmful and distressing content, including rape [36, 35], racist stereotypes [130], depictions of children [127], and derogatory taxonomies [182, 34, 128, 68]. The inclusion of such materials may further generate legal concerns [2]. We contend that these issues can be mitigated through the implementation of our recommendations.
> * Balancing resources between model development and data curation is a value-laden choice shaped by “social, political, and ethical values” [38]. While organizations readily invest significantly in model training [226, 334], compensation for data contributors often appears neglected [336, 335], disregarding that “most data represent or impact people” [378]. Remedial actions could be envisioned to bridge the gap between models developed with ethically curated data and those benefiting from expansive, nonconsensual crawled data. Reallocating research funds away from dominant data-hungry methods [38] would help to strike a balance between technological advancements and ethical imperatives. That being said, the granularity and comprehensiveness of our diversity recommendations could be adapted beyond evaluation contexts, particularly when employing “fairness without demographics” [209, 188, 133, 49] training approaches, reducing financial costs.

---

> ### Author Response · Authors · 2023-08-17
> **[Continued] Author Response**
>
> > I think this paper would be additionally enhanced and the recommendations more likely to be adopted if the authors could help HCCV researchers to make the transition from current ways of working to these much more ethical, but person time intensive, methods.
>
> We have now created an Appendix. In *Appendix A*, similar to existing AI ethics checklists [204, 224, 268, 355, 200, 89], we have created a checklist which translates each of our considerations and recommendations into action items for researchers and practitioners. Presented as a series of questions, mirroring the structure of our paper, these items are designed to stimulate discussions among data collection teams. The checklist is intended to be engaged with as a preliminary exercise before beginning data collection, promoting informed decision-making and minimizing risks. The checklist does not guarantee ethical compliance; rather, it functions as a catalyst for discussion and reflection.
>
> Our intention is that this addition aids data collection teams in transitioning, or at the very least, prompts contemplation about adopting more ethical practices.
>
>
>
> ---
>
> Let us know if these additions are satisfactory. Thank you for taking the time to review our paper and providing such positive feedback.

---

### Decision · Program_Chairs · 2023-09-22

**Decision:**

Accept (Oral)

**Comment:**

This paper is exceptionally written and presents important and timely work that is of relevance to the D&B Track. I agree with the reviewers that it warrants acceptance to the conference.